# Cumulative Exposure to Adverse Childhood Experience: Depressive Symptoms, Suicide Intensions and Suicide Plans among Senior High School Students in Nanchang City of China

**DOI:** 10.3390/ijerph17134718

**Published:** 2020-06-30

**Authors:** Zhihui Jia, Xiaotong Wen, Feiyu Chen, Hui Zhu, Can Li, Yixiang Lin, Xiaoxu Xie, Zhaokang Yuan

**Affiliations:** 1School of Public Health, Jiangxi Province Key Laboratory of Preventive Medicine, Nanchang University, Nanchang 330006, China; 401437618004@email.ncu.edu.cn (Z.J.); 406530517824@email.ncu.edu.cn (X.W.); linyixiang338@163.com (Y.L.); 2Center for Disease Control and Prevention, Dongxiang District, Fuzhou 331800, China; dxaids@163.com; 3Jiangxi Province Center for Disease Control and Prevention, Nanchang 330006, China; ncuzhuhui@126.com; 4Queen Mary School, Nanchang University, Nanchang 330006, China; can.li@se16.qmul.ac.uk; 5School of Public Health, Fujian Medical University, Fuzhou 350000, China

**Keywords:** adverse childhood experience, depressive symptoms, suicide intensions, suicide plans

## Abstract

This study tested relationships between different types of adverse childhood experiences (ACE) and depressive symptoms, suicide intensions, suicide plans and examines the cumulative effects of adverse childhood experience on depressive, suicide intentions and suicide plans among senior high school students. We conducted a survey among five senior high schools in Nanchang city, which were selected through stratified random cluster sampling. Among the 884 respondents, 409 were male (46.27%), and 475 were female (53.73%); the age ranged from 14 to 18. During the past 12 months, 199 (22.51%) students presented to depressive symptoms, 125 (14.14%) students had suicide intensions, 55 (6.22%) students had suicide plans. As ACE scores increased, there was an increase in the odds of (1) depressive symptoms—one ACE (adjusted odds ratio, AOR = 2.096, *p* < 0.001), two ACEs (AOR = 3.155, *p* < 0.001) and three to five ACEs (AOR = 9.707, *p* < 0.001); suicide intensions-1 ACE (AOR = 1.831, *p* = 0.011), two ACEs (AOR = 2.632, *p* = 0.002) and three to five ACEs (AOR = 10.836, *p* < 0.001); and (2) suicide plans—one ACE (AOR = 2.599, *p* < 0.001), two ACEs (AOR = 4.748, *p* < 0.001) and three to five ACEs (AOR = 22.660, *p* < 0.001). We should increase the awareness of adolescents who have had adverse childhood experience, especially those with multiple ACEs to prevent depression and suicide among senior high school students.

## 1. Introduction

Adverse childhood experiences (ACEs) can be defined as negative events experienced by an individual during his childhood, which will cause harm and threat to his psychological and physical health [1,2,3]. Negative events including such events as parental substance misuse and mental disorder, sexual abuse, physical abuse, emotional abuse, physical neglect, emotional neglect, familial death, parental separation, residential instability and witnessing violence in the home [4,5,6,7,8]. Adverse childhood experiences is a severe public health problem worldwide, imposing huge socioeconomic burdens and increasing affected individuals’ risk of mental disorders, including depressive symptoms and suicidal behaviors [9,10,11].

ACEs are very prevalent. Overall, 35.4% had one or more ACEs, according to the Minnesota Survey, which is a large (*n* = 105,759), statewide, anonymous survey of public high school students [12]. Based on the meta-analysis of 47 studies, the results estimated prevalence for Chinese child physical abuse was 36.6% [13]. In China, a systematic review estimated that among various types of abuse, physical abuse was the most prevalent (reported to be 26.6%), followed by neglect (26.0%), emotional abuse (19.6%) and sexual abuse (8.7%) [14].

In several studies, adverse childhood experiences have shown a significant association with depressive symptoms and suicidal behavior. Suicide is an ancient and cross-cultural theme and remains a public health problem worldwide [14]. Suicide ideations and plans were essential precursors of suicide and had a globally lifetime prevalence of 9.2% and 3.1%, respectively. Of American high school students, 17.2% had suicide ideation during the past 12 months according to youth risk behavior survey (YRBS) conduct by the Centers for Disease Control (CDC) of the United States [15]. According to the survey, 13.6% of high school students nationwide had suicide planning [15]. Over one million high school students in the United States are treated by a nurse or doctor annually for a suicide attempt [15]. According to the prior research findings described, we found that different types of adverse childhood experiences would increase the risk of suicidal behavior and have different extended effects on this behavior [16].

Children are often exposed to multiple forms of abuse and household dysfunction. There is evidence that cumulative exposure to adversities may have more significant impact on health outcomes than individual stressors. However, the co-occurrence of multiple ACEs is common, to the best of our knowledge, several prior studies had examined clusters of ACEs that can be identified as more closely related to suicidal behavior. However, adverse childhood experiences are known risk factors for suicidal behavior, we do not know the coexistence of these factors increases the risk. To best understand the impact of adversity in childhood, it is essential to consider this co-occurrence [17].

Recognition of the high co-occurrence of adversities led to the cumulative-risk approach widely used. This approach tallies the number of adversities experienced to create a risk score [18]. The cumulative-risk approach has been widely adopted and has proved useful for highlighting the public-health importance of adverse childhood experiences [17].

This study tested relationships between different types of ACEs and depressive, suicide intensions, suicide plans for senior high school students in Nanchang city, which is the capital city of Jiangxi province in China. Moreover, then, this study examines the cumulative exposure to adverse childhood experience and its influencing on depressive, suicide intentions and suicide plans among senior high school students.

## 2. Materials and Methods

### 2.1. Design

The study was designed to monitor health-related behaviors among high-school students. Our questionnaire was adapted from the youth risk behavior surveillance system (YRBSS), a public health project initiated by the US Centers for Disease Control and Prevention in 1989 to monitor major health hazards that cause deaths, illnesses and various social problems in high school students. Its questionnaire was used as the template for our study at five large public high schools in Nanchang city of Jiangxi Province, China. We modified the questionnaire so that it could be more suitable for this study, and the questionnaire was proved to be reliable and valid. The survey was conducted in May 2015, and the data collected were used in this study.

### 2.2. Participants

In May 2015, we surveyed among senior high school students attending five senior high schools in Nanchang city, and the schools were selected through stratified random cluster sampling. All the 34 senior high schools in Nanchang city were ranked from high to low according to their cutoff scores for the senior high school entrance examination. Sophomores and juniors in each school were included. The cumulative number of students in order reached 21,660 and a sampling interval of 4332 was produced by dividing the total number of students by five. Meanwhile, a random number of 2453, which was smaller than the sampling interval, was obtained from the random number table. The school in which the cumulative number of students corresponds to the random number is the first selected school. Then, a new random number was generated by adding this random number to the sampling interval. Similarly, the school with the corresponding number of students is the second selected school. In this way, five schools were selected. In each school, two grade 10 classes and two grade 11 classes were selected in the same way, and all the students in each class were investigated. A total of 950 questionnaires were distributed, and 884 questionnaires were collected. The effective rate of the questionnaire was 93.05%.

The students were instructed by previously trained investigators, who were postgraduate students majored in public health and preventive medicine at Nanchang University and the University of Nevada. They assisted the investigation process by explaining the purpose and content of the survey to the respondents, clarifying the specific requirements of the questionnaire, explaining the precautions, and dispelling the doubts of the respondent. Eventually, they obtained the trust of the respondents. Meanwhile, all students were informed that their responses would be anonymous as the questionnaires did not involve any collection of personal information. The questions asking about the students’ background would only be used to investigate the types of students completing this survey. The students were also informed about the importance of honesty and were told that their answers would not affect their grades at school. It took the students 45 min to complete the Chinese version questionnaires independently as required.

Among the 884 respondents, 409 were male (46.27%) and 475 were female (53.73%); the age ranged from 14 to 18 years, including 128 (14.48%) aged 15 years old or younger, 385 (43.55%) aged 16 years old, and 371 (41.97%) aged 17 years old or older.

### 2.3. Exposure Variable

#### 2.3.1. Adverse Childhood Experiences

This survey includes a measure of lifetime sexual abuse: “Have you ever been physically forced to have sexual intercourse when you did not want to?” Additionally, four variables were adapted from the behavioral risk factor surveillance system (BRFSS) ACE module [19,20,21] to assess the lifetime prevalence of (1) sexual abuse—“Have you ever been forced to have sexual intercourse when you did not want to?”; (2) physical abuse by an adult—“Have you ever been hit, beaten, kicked or physically hurt in any way by an adult? (Do not include being spanked for bad behavior)”; (3) household domestic violence—“Have you ever seen or heard adults in your home slap, hit, kick, punch or beat each other up?”; (4) household mental illness—“Have you ever lived with someone who was depressed, mentally ill or suicidal?”; and (5) household substance abuse—“Have you ever lived with someone who was a problem drinker or alcoholic or abused street or prescription drugs?” Responses to all ACE questions were dichotomized as yes versus no. The five ACE questions were summed to create a total ACE score (range 0–5). The ACE score was further categorized as 0, 1, 2 and 3–5 ACEs.

#### 2.3.2. Covariates

Demographic characteristics included sex, age and school type. The students were divided into three age groups, including 15 years old and younger, 16 years old, 17 years old, and and older. Moreover, school types were key senior high school, general senior high school and art senior high school.

### 2.4. Outcome Variable

#### 2.4.1. Depressive Symptoms

Depressive symptoms were assessed with a standardized question from the CDC YRBS: “During the past 12 months, did you ever feel so sad or hopeless almost every day for two weeks or more in a row that you stopped doing some usual activities?” The response choices were dichotomized as yes vs. no.

#### 2.4.2. Suicide Intensions

A standardized YRBS question was used to assess suicide intentions, “During the past 12 months, did you ever seriously consider attempting suicide?” Responses were dichotomized as yes versus no.

#### 2.4.3. Suicide Plans

Suicide attempts were assessed by asking students, “During the past 12 months, did you make a plan about how you would attempt suicide?” Responses were dichotomized as yes versus no.

### 2.5. Analyses

Epi Data 3.1 software (The Epi Data Association, Odense, Denmark) was used to enter data from two persons and two computers. All analyses were performed using the Statistic Package for Social Science 24.0 (IBM Corporation, Armonk, NY, USA). All statistical tests were two-sided with a significance level of *p* < 0.05. The analyses included three steps. First, descriptive statistics were used to summarize sociodemographic characteristics and ACEs of the sample. We used Pearson’s chi-square test to compare the incidence rate of depressive symptoms, the incidence rate of suicide intensions, the incidence rate of suicide plans with different sociodemographic factors and ACEs. Second, we assessed the relationship by multiple logistic regression models between sociodemographic factors, adverse childhood experience and depressive symptoms, suicide intensions, suicide plans of 884 Chinese senior high school students. Calculate different factors adjusted odds ratio (AOR) and its 95% confidence interval (95% CI). Statistical significance was determined according to the *P*-values. Finally, depressive symptoms, suicide intensions, suicide plans as predictors, we conduct multiple logistic regression models to explore the adverse childhood experience cumulative effect among Chinese senior high school students. AOR, 95% CI and *p*-value were reported.

### 2.6. Ethics Statement

This study was approved by the medical ethics committee of Nanchang University Institutional Review Board (2014-3-4) and conducted with the informed written consent of the school. Participants and their parents provided informed consent before data collection in a parents’ meeting. Students were informed that their participation in the study was voluntary, and their responses were anonymous.

## 3. Results

### 3.1. Describe Sociodemographic Characteristics and Adverse Childhood Experience of Sample

Table 1 first describes students’ sociodemographic characteristics and ACEs. Among the 884 respondents, 409 were male (46.27%), 475 were female (53.73%); the age ranged from 14 to 18 years, including 128 (14.48%) aged 15 years old or younger, 385 (43.55%) aged 16 years old, 371 (41.97%) aged 17 years old or older. There are 382 students (43.21%) in key senior high school, while 392 (44.35%) students in General senior high school and 110 (12.44%) students in art senior high school. Overall, 5.66% of the sample experienced sexual abuse, 10.41% of the sample experienced physical abuse, 21.49% of the sample experienced household domestic, 10.86% of the sample experienced household mental illness, and 4.75% of the sample experienced household substance use/abuse.

In this survey, 884 records were used. During the past 12 months, 199 (22.51%) students presented to depressive symptoms. During the past 12 months, 125 (14.14%) students had suicide intensions, 55 (6.22%) students had suicide plans.

### 3.2. Describe the Incidence rate of Depressive Symptoms, Suicide Intentions and Suicide Plans

As shown in Table 1, the incidence rate of depressive symptoms always higher than the incidence rate of suicide intensions among groups with different characteristics. Similarly, the incidence rate of suicide intensions always higher than the incidence rate of suicide plans among groups with different characteristics.

No statistically significant differences were detected regarding the gender, age among depressive symptoms, suicide intensions, suicide plans. However, the incidence rate of depressive symptoms was statistically significant difference among different school type groups (*p* < 0.05).

Table 1 also shows the unadjusted associations between different adverse childhood experience type groups and depressive symptoms, suicide intensions, suicide plans. The incidence rate of depressive symptoms, the incidence rate of suicide intensions, the incidence rate of suicide plans of students was statistically significant different between individual ACEs (all *p* < 0.01).

The incidence rate of depressive symptoms of students were statistically significant different among different ACE scores groups, such as sexual abuse (*χ^2^ =* 9.292, *p* = 0.002), physical abuse *(χ^2^ = 37.725, p <* 0.001), household domestic *(χ^2^ = 20.737, p <* 0.001), household mental illness *(χ^2^ = 39.848, p < 0.001*), household substance use/abuse *(χ^2^ = 13.056, p <* 0.001).

The incidence rate of suicide intensions of students was statistically significant difference among different ACE scores groups (all *p* < 0.05). The incidence rate of suicide plans of students was statistically significant difference among different ACE scores groups (all *p* < 0.01) (Table 1).

### 3.3. Associations of Adverse Childhood Experience and Depressive Symptoms, SUICIDE intensions, Suicide Plans

After controlling for all covariates, logistic regression models showed that higher odds of depressive symptoms were discovered among students who experienced physical abuse (OR = 2.621), household domestic (OR = 1.530) or household mental illness (OR = 2.789).

Logistic regression analyzed that higher odds of suicide intentions were found among students who experienced physical abuse (OR = 2.723) or household mental illness (OR = 3.153) under controlling for all covariates.

Logistic regression result displayed that after controlling for all covariates, higher odds of suicide plans were observed among students who experienced sexual abuse (OR = 2.894), physical abuse (OR = 3.189) or household mental illness (OR = 4.288). The results were shown in Table 2.

### 3.4. Associations of ACE Score and Depressive Symptoms, Suicide Intensions, Suicide Plans

After controlling for all covariates, logistic regression models showed that much higher odds of depressive symptoms were discovered among students whose ACE score were much higher. Suicide intentions and planning are similar. As ACE scores increased, there was an increase in the odds of (1) depressive symptoms-1 ACE (AOR = 2.096, *p* < 0.001), 2 ACEs (AOR = 3.155, *p* < 0.001) and 3–5 ACEs (AOR = 9.707, *p* < 0.001); suicide intensions-1 ACE (AOR = 1.831, *p* = 0.011), 2 ACEs (AOR = 2.632, *p* = 0.002) and 3–5 ACEs (AOR = 10.836, *p* < 0.001); and (2) suicide plans-1 ACE (AOR = 2.599, *p* < 0.001), 2 ACEs (AOR = 4.748, *p* < 0.001) and 3–5 ACEs (AOR = 22.660, *p* < 0.001) (Table 3). Figure 1 showed that the cumulative effect of ACEs on depressive symptoms, suicide intensions, suicide plan.

## 4. Discussion

This study examined the relationship between each ACEs, including sexual abuse, physical abuse, domestic violence, family mental illness, family substance abuse and three outcome variables: depression, suicide intentions and suicide plans. However, we found that 34.16% of the sample self-reported at least exposure to one type of ACEs, which was low compared with the previous studies. It is possible that only five variables were used as the ACE index in our study, which referred to the behavioral risk factor surveillance system (BRFSS). Previous studies may take more variables as ACE index, such as Remy M selected parental divorce [22], and Tracie O. Afifi utilized expanded ACE index that included spanking [23].

Of students, 22.5% reported depressive symptoms in this study, which is slightly lower than 24.3% among Chinese adolescents (2018 years) [24]. Relevant studies have shown that the prevalence of depressive symptoms among Chinese adolescents was very likely higher than those in other countries [25,26,27]. Tremendous academic pressure may account for the high prevalence of depression in Chinese adolescents [28]. Our study also found that differences in depressive symptoms among students in three types of schools were statistically significant, which may be caused by the differences in academic pressure. In addition, interpersonal pressure and specific cultural factors may also lead to depression [24].

Suicide intentions and plans were reported at 6.9% and 4.0% in a recent study of Spanish adolescents [4]. Similarly, it was also found that at least 4.2% of adolescents planned to suicide in a study among 15,191 sample adolescents in the United States [8]. Suicide is a leading cause to death for adolescents in Australia [29]. However, only 5.5% and 2.9% suicide behaviors were also reported in the Australian survey of 12- to 17-year-olds [30]. Suicide intentions and plans accounted for17.7% and 7.3% in mainland China, and there are 14.1% attempted suicide and 6.2% planed among adolescents in this study, which is higher than other countries [31]. One important reason is that there are more than 68 million left-behind children in China [32], and they are more susceptible to being victimized by discrimination, humiliation and bullying [33], causing suicide [34,35]. Moreover, academic pressure accounts for the first place (45.5%) among the causes of suicide in Chinese adolescents.

Adolescents are key objects of protection in the field of public health. The results showed that ACEs were important determinants of chronic physical problems in adolescents [36,37,38]. Our study aimed to provide suggestions on how to promote adolescent health better. However, previous studies used retrospective design to examine the relationship between adverse childhood experiences and outcomes in adulthood, which interval time was too long to recall accurately [39]. In other words, our study surveyed high school students and reduced error and bias.

The results indicated that each type of ACEs was significantly associated with depressive symptoms, suicide intentions and suicide plans. According to statistics, one-fifth of girls in the world are subject to sexual abuse [40]. Our tentative results found that depressive symptoms, suicide intentions, and suicide plans were associated with sexual abuse. In adjusted models, people who reported sexual and physical abuse had an increased risk of suicide plans, which is consistent with an Israeli study [41]. There was also a correlation between sexual abuse and depressive symptoms. Physical and sexual abuse victims in childhood were at increased risk of depression, anxiety, and post-traumatic stress disorder (PTSD) in adulthood [42]. Children suffered in the process of mature experience, which is the critical risk factors of depression [43]. Furthermore, previous research showed that male who experienced sexual abuse in childhood had higher levels of depressive symptoms after 50 years old, and the exposure time of the study samples’ depression symptoms may be delayed [44].

Physical abuse also is a risk factor for depressive symptoms, suicide intentions and suicide plans. Compared to students with no physical abuse, individuals who suffered physical abuse had 2.73, 2.72, 3.19 times increased odds of reporting depressive symptoms, suicide intentions and suicide plans. Inbar Kremer also said, such as physical abuse victims more attractive to death [41]. One study estimated that a quarter of children had experienced physical abuse worldwide [45]. Chromosomal and functional brain damage [46,47,48], proximal behavioral problems [49] and misconduct [42] are all associated with physical abuse throughout life [50]. Compared with emotional abuse, it is relatively easy to identify physical abuse and sexual abuse in the clinical environment because the latter two have more apparent signs [51]. Therefore, we should give full play to the important role of controlling physical abuse and sexual abuse in reducing bad behaviors in high school students.

Family mental illness is a risk factor for depressive symptoms, suicide intentions and suicide plans. Compared to students with no FMI, individuals reporting the ACE had 2.79 ime-increased odds of reporting depressive symptoms, 3.15 times increased odds of attempting suicide, 4.29 times increased odds of reporting suicide plans during childhood. A classic review has found that parental psychopathology, such as substance abuse, depression or antisocial behavior, is the most important predictor of adolescent suicide [13].

Household violence is a risk factor for depression, and high school students who experienced domestic violence had a 1.53-fold increase in depressive symptoms. No country is immune to the problem of violence, including domestic violence, the consequences of which can be devastating [52]. There are gender differences in sexual abuse, physical abuse and household violence in childhood and female are vulnerable groups [53], which reminds us to pay more attention to the mental health of women.

In addition to all this, our results align with previous research to identify that a general dose–response relationship between ACE score and adult behavior results [54,55,56]. As the ACE score increased, the odds of experiencing depressive symptoms, suicide attempts and suicide plans in adulthood also increased. Take depression symptoms as an example, compared to individuals with no ACEs, students reporting one ACE had 2.10 times increased odds, two ACEs had 3.16 times increased odds and three or more ACEs had 9.70 times increased odds, Which are consistent with those of previous studies on the correlation between ACES and adverse behavior outcomes [57].

## 5. Limitations

The severity, frequency and time of ACE factors were not included in the study. In addition, our study concluded that as ACEs scores increased, the probability of occurrence of the outcomes went up, and each increment of adversity would increase the occurrence. Nevertheless, the relationship between ACEs and ACEs, whether the occurrence of one kind of ACEs increases the probability of another, for example, Michael T proposed that once per young person is exposed to one kind of ACEs, the probability of another kind of ACEs increased 1286 times [58], was not designed in this study. It has to be said that the scope of our study ACE indicators is not wide enough, the exposure and results may be low, and there is an outcome bias. Self-reported tests of children’s adverse experiences and outcome variables may bias and reduce actual prevalence.

## 6. Conclusions

To our knowledge, this is one of the few studies to explore the relationships between adolescents’ adverse experiences and their psychosocial behavioral health in a representative sample of senior high school students. In this study, we compared the data with other countries and found that the depressive symptoms and suicidal behaviors of Chinese adolescents were relatively severe. On this basis, we analyzed the effects of each ACE and the cumulative ACE scores on the health outcomes of high school students. First, we found many adverse experiences are risk factors for depression and suicidal behaviors in adolescents; second, we confirmed that there was a dose–response relationship between ACEs and depressive symptoms and suicide intentions and plans. Therefore, we should focus on starting at the source to prevent depression and suicide among high school students, increasing the attention to adolescents who had adverse experiences, especially those with the multiple, and playing an important role in controlling key risk factors in the protection of adolescent health. Further studies will be necessary to explore the influence of different characteristics of ACEs on outcomes and study the relationship between ACEs.

## Figures and Tables

**Figure 1 ijerph-17-04718-f001:**
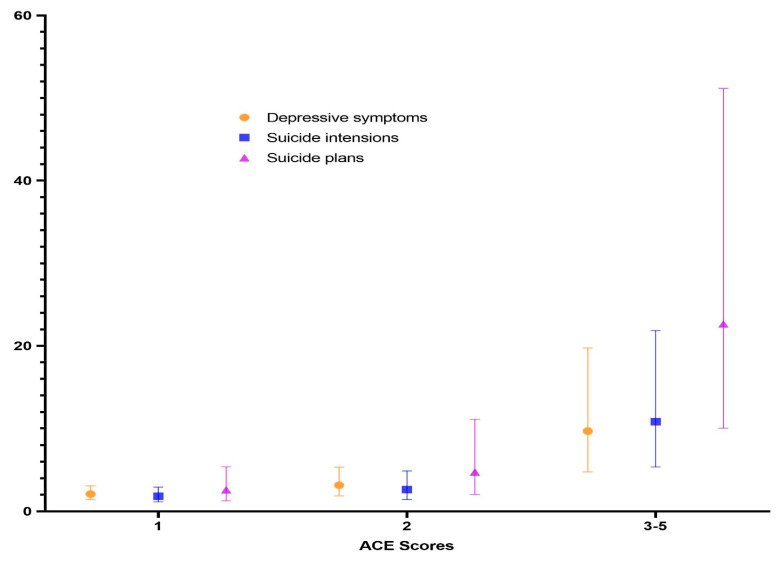
Cumulative effect of ACEs on depressive symptoms, suicide intensions and suicide plans of 884 Chinese high school students.

**Table 1 ijerph-17-04718-t001:** Unadjusted associations of adverse childhood experience, sociodemographic characteristics and depressive symptoms, suicide intensions, suicide plan of 884 Chinese high school students, 2015.

Variables	Total (*N* = 884)	Depressive Symptoms *n* (%)	Suicide Intensions *n* (%)	Suicide Plans *n* (%)	*p*-Value
Sex					
Female	475 (53.73)	110 (23.16)	70 (14.74)	26 (5.47)	
Male	409 (46.27)	89 (21.76)	55 (13.45)	29 (7.09)	
Age (year)					
≤15	128 (14.48)	23 (17.97)	13 (10.16)	8 (6.25)	
16	385 (43.55)	93 (24.16)	61 (15.84)	21 (5.45)	
≥17	371 (41.97)	83 (22.37)	51 (13.75)	26 (7.01)	
School					
Key senior high school	382 (43.21)	100 (26.18)	62 (16.23)	30 (7.85)	A1
General senior high school	392 (44.35)	72 (18.37)	45 (11.48)	16 (4.08)	
Art senior high school	110 (12.44)	27 (24.55)	18 (16.36)	9 (8.18)	
Sexual abuse					A2 B1 C3
No	834 (94.34)	179 (21.46)	112 (13.43)	43 (5.16)	
Yes	50 (5.66)	20 (40.00)	37 (74.00)	12 (24.00)	
Physical abuse					A3 B3 C3
No	792 (89.59)	155 (19.57)	94 (11.87)	35 (4.42)	
Yes	92 (10.41)	44 (47.83)	31 (33.70)	20 (21.74)	
Household domestic violence					A3 B3 C2
No	694 (78.51)	133 (19.16)	83 (11.96)	35 (5.04)	
Yes	190 (21.49)	66 (34.74)	42 (22.11)	20 (10.53)	
Household mental illness					A3 B3 C3
No	788 (89.14)	153 (19.42)	91 (11.55)	32 (4.06)	
Yes	96 (10.86)	46 (47.92)	34 (35.42)	23 (23.96)	
Household substance use/abuse					A3 B3 C3
No	842 (95.25)	180 (21.38)	111 (13.18)	43 (5.11)	
Yes	42 (4.75)	19 (45.24)	14 (33.33)	12 (28.57)	
ACE Score					A3 B3 C3
0	582 (65.84)	93 (15.98)	57 (9.79)	17 (2.92)	
1	193 (21.83)	55 (28.50)	32 (16.58)	14 (7.25)	
2	72 (8.14)	27 (37.50)	16 (22.22)	9 (12.50)	
3–5	37 (4.19)	24 (64.86)	20 (54.05)	15 (40.54)	
Total	884 (100.00)	199 (22.51)	125 (14.14)	55 (6.22)	–

Note: AOR = adjusted odds ratio; CI = confidence interval; ref = reference. A: depressive symptoms, A1 < 0.05; A2 < 0.01; A3 < 0.001; B: suicide intensions, B1 < 0.05; B2 < 0.01; B3 < 0.001; C: suicide plan, C1 < 0.05; C2 < 0.01; C3 < 0.001.

**Table 2 ijerph-17-04718-t002:** Adjusted associations of adverse childhood experience and sociodemographic characteristics with depressive symptoms, suicide intensions and suicide plans of 884 Chinese high school students, 2015 AOR (95% CI).

Variables	Depressive Symptoms	Suicide Intensions	Suicide Plans	*p*-Value
Sex				
Female (ref)	1	1	1	
Male	0.779 (0.553,1.097)	0.743 (0.491,1.122)	0.920 (0.493,1.715)	
Age				
≤15 year (ref)	1	1	1	
16 year	1.607 (0.936,2.759)	1.941 (0.989,3.807)	0.956 (0.387,2.360)	
≥17 year	1.380 (0.798,2.386)	1.489 (0.752,2.950)	1.036 (0.425,2.522)	
School				
Key senior high school (ref)	1	1	1	
General senior high school	0.719 (0.501,1.031)	0.779 (0.504,1.202)	0.607 (0.311,1.186)	
Art senior high school	0.969 (0.574,1.637)	1.065 (0.577,1.968)	1.061 (0.448,2.516)	
Sexual abuse				
No (ref)	1	1	1	
Yes	1.565 (0.802,3.054)	1.334 (0.623,2.858)	2.894 (1.210,6.924)	C1
Physical abuse				
No (ref)	1	1	1	
Yes	2.621 (1.603,4.285)	2.723 (1.581,4.689)	3.189 (1.549,6.567)	A3 B3 C2
Household domestic violence				
No (ref)	1	1	1	
Yes	1.530 (1.034,2.262)	1.347 (0.849,2.136)	0.974 (0.489,1.938)	A1
Household mental illness				
No (ref)	1	1	1	
Yes	2.789 (1.703,4.567)	3.153 (1.839,5.404)	4.288 (2.097,8.766)	A3 B3 C2
Household substance use/abuse				
No (ref)	1	1	1	
Yes	1.202 (0.570,2.536)	1.231 (0.551,2.754)	1.975 (0.761,5.125)	

Note: AOR = adjusted odds ratio; CI = confidence interval; ref = reference. A: depressive symptoms, A1 < 0.05; A2 < 0.01; A3 < 0.001; B: suicide intensions, B1 < 0.05; B2 < 0.01; B3 < 0.001; C: suicide plans, C1 < 0.05; C2 < 0.01; C3 < 0.001.

**Table 3 ijerph-17-04718-t003:** Cumulative effect of ACEs on depressive symptoms, suicide intensions and suicide plans of 884 Chinese high school students—2015.

ACE Score	Depressive Symptoms		Suicide Intensions		Suicide Plans	
	AOR (95% CI)	*p*-Value	AOR (95% CI)	*p*-Value	AOR (95% CI)	*p*-Value
0 (ref)	1	–	1	–	1	–
1	2.096 (1.429, 3.074)	<0.001	1.831 (1.147,2.922)	0.011	2.599 (1.256,5.378)	0.010
2	3.155 (1.864, 5.339)	<0.001	2.632 (1.417,4.888)	0.002	4.748 (2.032,11.096)	<0.001
3–5	9.707 (4.770,19.753)	<0.001	10.836 (5.370,21.864)	<0.001	22.660 (10.035,51.170)	<0.001

AOR = adjusted odds ratio; CI = confidence interval; ref = reference.

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
