# Peer review of "Cumulative Exposure to Adverse Childhood Experience: Depressive Symptoms, Suicide Intensions and Suicide Plans among Senior High School Students in Nanchang City of China"

_ijerph, 2020, doi:10.3390/ijerph17134718_

Round 1

Reviewer 1 Report

This a well conceived and designed study

The English is not good enough and I found difficulty in following the article at times

Although comparisons are made with studies in other countries the differences between findings are not discussed. What is different in China and are there tentative explanations for these differences

Author Response

Dear reviewer,

I very much appreciate the thorough and constructive criticism given by reviewers. I have made some changes according to the comments and suggestions of the reviewers. The text has been revised and edited for language so as to present a more clear and compelling narrative. These changes can improve the manuscript, I hope.

Reviewer: Comments and Suggestions for Authors

Review for ijerph-840022 “Cumulative Exposure to Adverse Childhood Experience :Depressive Symptoms, Suicide Intensions and Suicide Plans among Senior High School Students in Nanchang City of China”.

This a well conceived and designed study.

ü  The English is not good enough and I found difficulty in following the article at times.

Response:Thank you for your suggestions. We check the grammatical errors and possible English errors carefully.

ü  Although comparisons are made with studies in other countries the differences between findings are not discussed. What is different in China and are there tentative explanations for these differences.

Response:Thank you for your suggestions. We search and find more papers to discuss some results. Line 253-259, Page 8; Line 264-270,Page 8.

Reviewer 2 Report

Thank you for inviting me to review this manuscript reporting on the results of a survey assessing the associations between adverse childhood events (ACE) and mental health-related outcomes in students. I have only few questions.

The survey inquired about several ACEs. Have you looked at the correlations between the ACEs and have the correlations been taken into account in the analysis? How? 

The survey was conducted in 2015. How come authors have waited five years before preparing or submitting this manuscript? 

Author Response

Dear reviewer,

I very much appreciate the thorough and constructive criticism given by reviewers. I have made some changes according to the comments and suggestions of the reviewers. The text has been revised and edited for language so as to present a more clear and compelling narrative. These changes can improve the manuscript, I hope.

Reviewer: Comments and Suggestions for Authors

Review for ijerph-840022 “Cumulative Exposure to Adverse Childhood Experience :Depressive Symptoms, Suicide Intensions and Suicide Plans among Senior High School Students in Nanchang City of China”.

Point 1:The survey inquired about several ACEs. Have you looked at the correlations between the ACEs and have the correlations been taken into account in the analysis? How?

Response1:Thank you for your suggestions. We focused on the comprehensive analysis of each ACE and cumulative ACEs influence on depression and suicidal behaviors, so this paper did not mention the relationship of ACEs. We have marked this in "Limitations"(Line 322-325, Page 8), and we will focus on the study of ACEs relationship in the next research.

Point 1:The survey was conducted in 2015. How come authors have waited five years before preparing or submitting this manuscript?

Response2:Thank you for your suggestions. We have been actively exploring adverse childhood experiences recently. As far as we know, in China or even in other countries,there are few studies that analyze the influence of ACE on depression and suicidal behaviors among adolescents.  In this survey, high school students in Nanchang city were selected as samples, both the questionnaires and the investigators went through strict examination and training when we conducted. We believe that these data are still representative, authentic up to now, and have reference value. So we wrote and submitted this manuscript.

Reviewer 3 Report

Nicely done but study did not seem to demonstrate anything new to the field.  We all know the more ACEs the more depression and suicidal ideation and behavior.  So novel approach to getting to the same place as other have before.  I felt the conclusions were light and could be improved, as were the limitations.  I felt there wasn't enough discussion of why some results (e.g. depression ratings) were so low comparatively.  Overall, fine study.

Author Response

Dear reviewer,

I very much appreciate the thorough and constructive criticism given by reviewers. I have made some changes according to the comments and suggestions of the reviewers. The text has been revised and edited for language so as to present a more clear and compelling narrative. These changes can improve the manuscript, I hope.

Reviewer: Comments and Suggestions for Authors

Review for ijerph-840022 “Cumulative Exposure to Adverse Childhood Experience :Depressive Symptoms, Suicide Intensions and Suicide Plans among Senior High School Students in Nanchang City of China”.

Point: Nicely done but study did not seem to demonstrate anything new to the field.  We all know the more ACEs the more depression and suicidal ideation and behavior.  So novel approach to getting to the same place as other have before.  I felt the conclusions were light and could be improved, as were the limitations.  I felt there wasn't enough discussion of why some results (e.g. depression ratings) were so low comparatively.  Overall, fine study.

Response : Thank you for your suggestions. This is a paper with adolescents psychological and behavioral health as the research topic. Previous research mostly studied the relationship between ACEs in childhood and health outcomes in adulthood. However, the purpose of our study essentially is to find the influencing factors of adolescent depression and suicide behaviors from the perspective of ACE, so as to provide certain scientific basis for the intervention of adolescents physical and mental health, which is innovative to some extent.

Thank you for your suggestions. We add the conclusions. Line 332-338, Page 9.

Thank you for your suggestions again. We search and find more papers to discuss some results. Line 253-259, Page 8; Line 264-270, Page 8.
